# Recent Advances in the Equal Channel Angular Pressing of Metallic Materials

**Lang Cui [1], Shengmin Shao [2], Haitao Wang [3], Guoqing Zhang [4], Zejia Zhao [1],\* and Chunyang Zhao [4],\***

[1] Institute of Semiconductor Manufacturing Research, College of Mechatronics and Control Engineering, Shenzhen University, Shenzhen 518060, China

[2] China Electronics Technology Group Corporation 52nd Research Institute, Hangzhou 310061, China

[3] School of Mechanical and Electrical Engineering, Shenzhen Polytechnic, Shenzhen 518055, China

[4] Shenzhen Key Laboratory of High Performance Nontraditional Manufacturing, Guangdong Provincial Key Laboratory of Electromagnetic Control and Intelligent Robots, College of Mechatronics and Control Engineering, Shenzhen University, Shenzhen 518060, China

\* Correspondence: zhaozejia@szu.edu.cn (Z.Z.); zcy724317@163.com (C.Z.)

**Abstract:** Applications of a metallic material highly depend on its mechanical properties, which greatly depend on the material's grain sizes. Reducing grain sizes by severe plastic deformation is one of the efficient approaches to enhance the mechanical properties of a metallic material. In this paper, severe plastic deformation of equal channel angular pressing (ECAP) will be reviewed to illustrate its effects on the grain refinement of some common metallic materials such as titanium alloys, aluminum alloys, and magnesium alloys. In the ECAP process, the materials can be processed severely and repeatedly in a designed ECAP mold to accumulate a large amount of plastic strain. Ultrafine grains with diameters of submicron meters or even nanometers can be achieved through severe plastic deformation of the ECAP. In detail, this paper will give state-of-the-art details about the influences of ECAP processing parameters such as passes, temperature, and routes on the evolution of the microstructure of metallic materials. The evolution of grain sizes, grain boundaries, and phases of different metallic materials during the ECAP process are also analyzed. Besides, the plastic deformation mechanism during the ECAP process is discussed from the perspectives of dislocation slipping and twinning.

**Keywords:** grain size; ECAP; metallic materials; processing parameters; deformation mechanism

## 1. Introduction

The mechanical properties of metallic materials depend to a large extent on the grain size, and reducing the grain size has great potential to improve the mechanical properties of metals and their alloys. The yield strength $\sigma_y$ of the material and the grain diameter d conform to the Hall–Petch relationship ($\sigma_y = \sigma_0 + k_y d^{-\frac{1}{2}}$) [1,2], where $\sigma_0$ and $k_y$ are constants about the material. It can be seen from the relationship that the smaller the grain size, the higher the yield strength of the material. For example, Magnesium and its alloys have high biosafety and can be used in the medical field. Reducing the grain size can improve their mechanical properties and deformability, showing great application potential in the preparation of medical devices and medical materials [3]. Furthermore, a number of studies have shown that titanium and its alloys with ultrafine grains have superior strength, plasticity, fatigue resistance, corrosion resistance, and many other excellent properties, which have been widely used in the fields of aerospace, biomedicine and chemical machinery [4–7].

Many methods have been applied to grain refinement nowadays, such as traditional thermomechanical and metal forming processes, ultrasonic vibration, and severe plastic deformation [8,9]. Ultrafine and nanoscale grains are difficult to obtain by conventional

thermomechanical and metal forming processes, and ultrasonic vibration also has some limitations due to the corrosion and reactivity of the radiator under high-temperature conditions [10,11]. Among these methods, severe plastic deformation (SPD) was reported to be an efficient and low-cost technique that has several advantages over other approaches to refine the grains to micro or nano meters of metallic materials, such as wide applicability of workpiece shapes, suitable for handling bulk materials [10] and high levels of strain [12]. Commonly used severe plastic deformation methods include high-pressure torsion (HPT), equal channel angular pressing (ECAP), twist channel angular pressing (TCAP), twist channel multi-angular pressing (TCMAP), cyclic extrusion and compression (CEC), accumulative rolling bonding(ARB), etc. [12–16].

The ECAP technique can prepare large-scale ultrafine grain bulk materials without changing the workpiece size and area of the material, which has great application potential in comparison to other SPD methods [14,17]. Many scholars have used this method to alter the microstructure and improve the mechanical properties of non-ferrous metals including Al, Ti, Mg, Cu, and their alloys [17,18]. Previous studies have shown that the improvement of material properties by ECAP is affected by many factors, such as passes, temperature, routes, back pressures, channel angle, etc., and the grain size can be effectively reduced by controlling the processing parameters [14,17,19,20]. The strain distribution of aluminum after different channel angles, curvature angles, and passes was simulated by the finite element method. The strain uniformity was quantitatively studied by using a non-uniformity index and standard deviation, and the best suggestions for die parameters were given. When the channel angle is obtuse, the effective strain decreases with the increase of the angle; When the channel angle is acute, the deformation is more uneven and complex; When the channel angle is a right angle, the effective change is uniform [20,21]. Figure 1 schematically shows the ECAP process and schematic diagram of different routes. In addition, the addition of subsequent different treatments based on the study of ECAP also affects the properties of metal materials. For example, studies have shown that the strength of the material is further improved after ECAP annealing, but the plasticity is almost unchanged. The deformation microstructure and phase structure of the brass before and after heat treatment were also not changed [22,23]. The ECAP technology shows great potential for industrial application in grain refinement, making great contributions to the development of various fields.

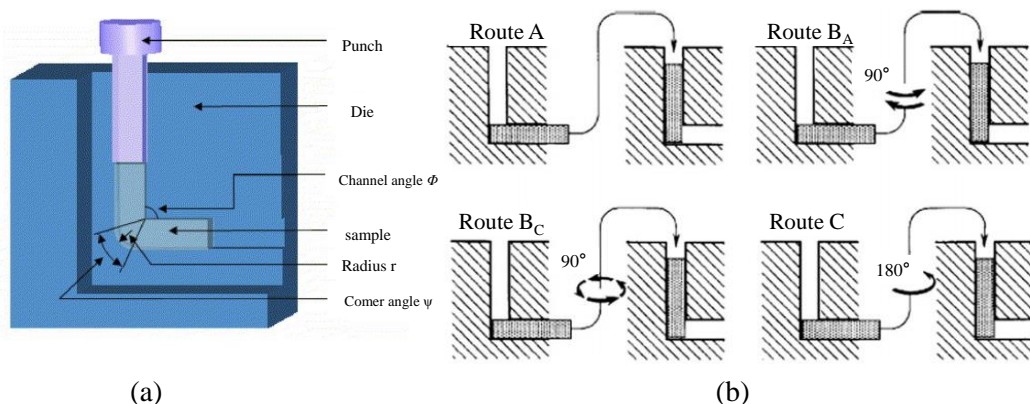

**Figure 1.** (**a**) Working principle of ECAP [24]. Reproduced with permission from Elsevier B.V. (**b**) Schematic diagram of different routes [17]. Reproduced with permission from Elsevier Ltd.

Even there are several reviews about the ECAP, most of the existing studies focus on summarizing the effects of ECAP parameters on material properties, and little research was conducted on reviewing the microstructure evolution after the ECAP from perspectives of various grain sizes, grain boundaries and phase compositions. Therefore, In this review, the microstructure evolution of the metallic materials induced by the ECAP is to

be analyzed and discussed from the views of grain size evolution, grain boundaries, and phase evolution, as summarized in Figure 2.

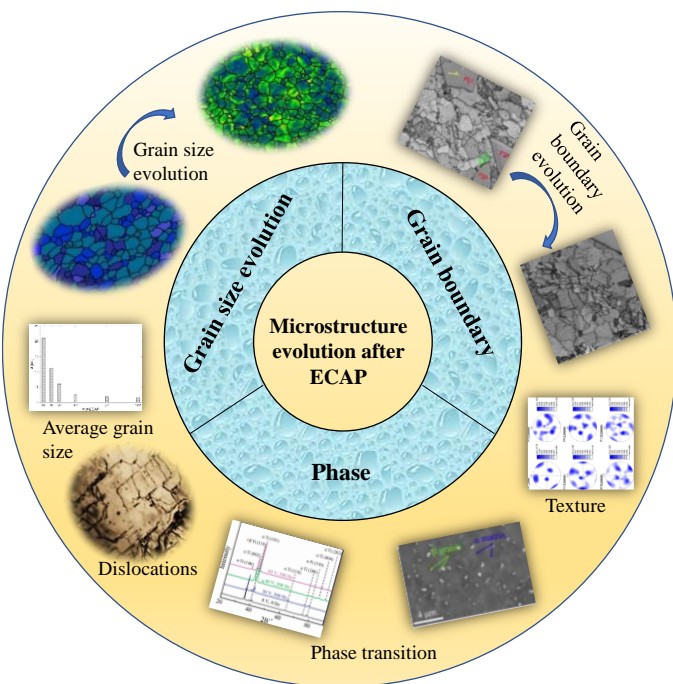

**Figure 2.** Overview map [25–31]. Reproduced with permission from Elsevier B.V. Copyright 2014. Reproduced with permission from Elsevier Editor Ltd. Reproduced with permission from Springer Nature. Reproduced with permission from MDPI. Reproduced with permission from Elsevier B.V. Reproduced with permission from Elsevier Ltd. Reproduced with permission from Elsevier Ltd.

## 2. Microstructure Evolution after ECAP

The microstructure can strongly influence the properties of materials, which primarily includes grain sizes, grain boundaries and phases. In this section, the effects of ECAP on the microstructural evolution of metallic materials will be discussed in detail by analyzing the evolution of grain sizes, grain boundaries, and phases. The metallic materials mainly include Ti alloys, Mg alloys and Al alloys.

### 2.1. Grain Size Evolution

To demonstrate the grain refinement induced by the ECAP, the grain size is divided into three scales, i.e., from 1 to 10 μm, from 0.1 to 1 μm, and below 0.1 μm. The processing parameters in the ECAP process can directly affect the microstructure of the material and its grain size. Different processing parameters have different effects on the microstructure of the material, which contributes to various mechanical properties. The contributory parameters affecting the grain size evolution primarily include ECAP passes, routes, and temperature. Next, the grains of different size ranges will be discussed mainly from the above factors.

#### 2.1.1. Grain Size from 1 to 10 μm

ECAP pass is one of the primary factors that affect the evolution of grain sizes. Zhao et al. preheated the sample to 923 K for 5 min before extrusion and carried out non-isothermal ECAP with the extrusion rate of 20 mm/s and B$_C$ route. They found that the average grain size of titanium alloys decreased from 15–20 μm to 5–10 μm and from 30 μm to 10 μm after four passes of ECAP, respectively [32,33]. Jahadi et al. reported that the grain size of magnesium alloys was reduced from the initial 20.4 μm to 3.9 μm after four passes of ECAP at 275 °C [34]. Matsubara et al. found that the size of magnesium alloys decreased by almost the same proportion after four passes at 300 °C, from the initial 50–10 μm [35].

It should be noted that the initial grain size refers to the grain size of the metallic materials before the ECAP processing. In comparison, it is found that the grain size of the material decreases to 1–10 μm under similar conditions. Figure 3 shows the microstructures of titanium and magnesium alloys after four passes of ECAP.

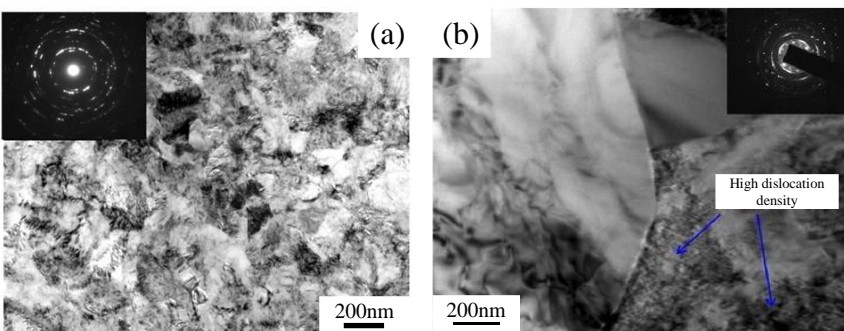

**Figure 3.** TEM image of the material after four passes: (**a**) titanium alloy [32]. Reproduced with permission from MDPI. (**b**) magnesium alloy [36]. Reproduced with permission from Springer Nature.

Figure 4 shows the EBSD images of the commercially pure aluminum after different passes, and the microstructure variation with ECAP passes could be clearly seen from the images. Kawasaki et al. studied the microstructure of commercially pure Al by selecting the Bc route and pressing rate of 7 mm/s at room temperature. When other conditions are the same, the grain size will be refined to different degrees after different passes. The grain size was reduced to 1.8 μm after two passes and 1.4 μm after three passes. With the increase of pass number, the average grain size was measured to be 1.3 μm after four passes of ECAP and 1.1 μm after eight passes of ECAP [25]. Terhune et al. obtained similar data at room temperature and Bc routes, with a grain size of 1.2 μm in four passes and 1.0 μm in twelve passes [37]. Generally speaking, the degree of grain size reduction becomes smaller and smaller as the pass increases. However, when the number of passes increased to twelve, the grain size was measured to be 1.2 μm, which increased somewhat for the commercially pure Al [25]. A similar situation is also found in the pure Ti and Al alloy, which proves that the grain size of the material will hardly change or even become coarse after a certain number of passes [38,39].

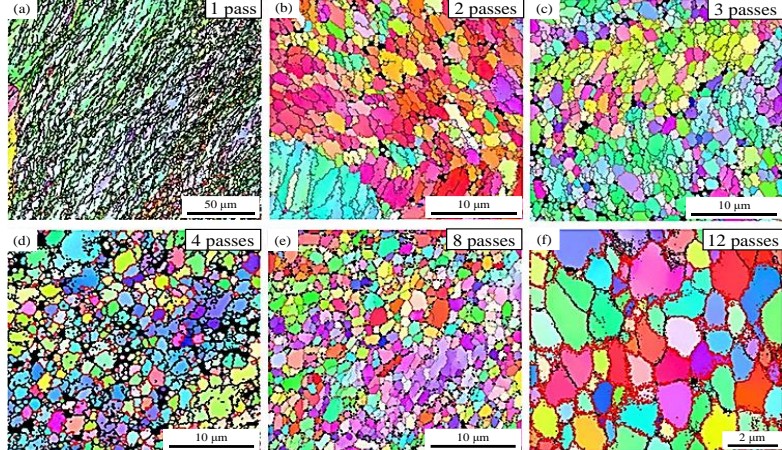

**Figure 4.** OIM images under different passes [25]: (**a**) one pass; (**b**) two passes; (**c**) three passes; (**d**) four passes; (**e**) eight passes; (**f**) twelve passes. Reproduced with permission from Elsevier B.V.

The grain size of the metal matrix composites (MMCs) can also be reduced by the ECAP process. Ramu and Bauri prepared the aluminum matrix composites with SiCp volume fractions of five and ten by the stirring casting method. After two passes of ECAP,

the grain size of the Al/5SiCp composite was reduced to 8 μm, while the Al/10SiCp could only undergo one ECAP (up to 16 μm), and then cracks appeared. This is because higher SiC content affects the strain rate sensitivity and the degree of strain hardening [40].

The ECAP route also affects the change in grain sizes. Kocich et al. showed that the grain size of commercially pure aluminum decreased to 8 μm and 7.4 μm after two-pass ECAP under the A and Bc routes, respectively [41].Gottstein et al. reported that the average grain size of pure Mg reached 6–8 μm after four passes under the A, Bc, and C routes [42]. Tong et al. found that the average grain size of Mg alloys decreased from 4.3 μm to 1.3 μm under the A route, and the grain size decreased to less than 1 μm under both the Bc and C routes [43]. El-Danaf et al. performed eight passes of ECAP on pure Al under the Bc route and the C route, and the measured average grain sizes were 1.1 μm and 2.9 μm, respectively [44]. Tong et al. also found that the ECAP process of the A and Bc routes increased the material yield strength due to the grain refinement strengthening effect and the texture softening competition, while the C route induced a reduction of the yield strength [43]. The microstructures of Mg under different routes are shown in Figure 5. Based on previous studies, the Bc route is more likely to contribute to the best grain refinement effect.

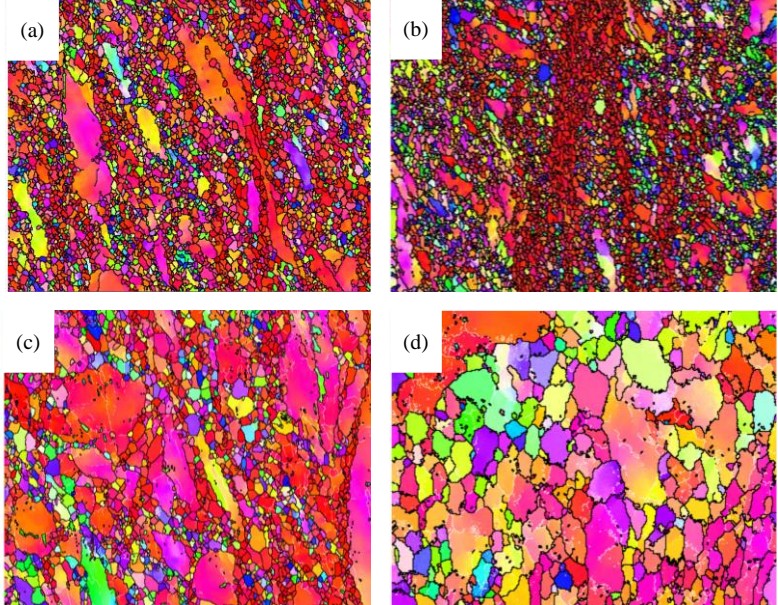

**Figure 5.** Microstructure of magnesium under different routes [45]: (**a**) A route; (**b**) C route; (**c**) Bc route after 1 pass; (**d**) Bc route after four passes. Reproduced with permission from MDPI.

Some special routes were also proposed to refine the grain sizes. Gajanan et al. compared the microstructure and mechanical properties of Mg alloys after passing through the Bc and R (inversions per pass) routes at 390 isothermal conditions. The research shows that the R route has a better microstructure refinement effect; grain refinement and secondary phase distribution are more uniform, and the mechanical properties are better than those processed by the Bc route [46]. Li et.al. developed a new route $B_{135}$ by defining 135° clockwise rotations of the samples per pass and ECAP treatment at a temperature of 573 K. After four passes, the grain sizes under the $B_{135}$ route and the Bc route are 2.36 μm and 2.82 μm, respectively. Compared with the Bc route, it is found that the new route $B_{135}$ has a better grain refinement effect [47]. Figure 6 shows the evolution of grain sizes after different ECAP routes.

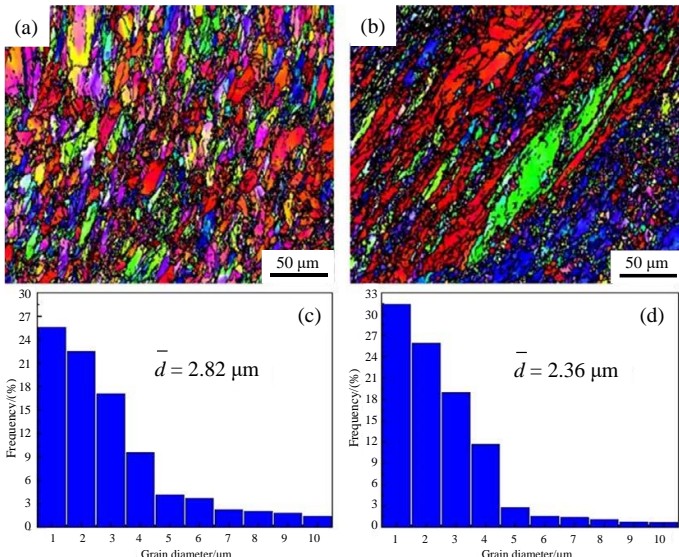

**Figure 6.** EBSD images under different routes [47]: (**a**) B_C route; (**b**) B_135 route; (**c,d**) are the corresponding grain size histograms, respectively. Reproduced with permission from IOP Publishing Ltd.

Processing temperature also has some effect on grain size during ECAP. Lin et al. found that the average grain size of AZ31 Mg alloy reached about 2.5 μm after one pass of ECAP at 300 °C [48]. Kim and Jeong reported that the average grain size of AZ31 alloy was reduced from 24 μm to 4.8 μm, 3.2 μm and 2.2 μm after six passes of ECAP at three different temperatures (553 K, 523 K, 493 K), respectively [49]. The average grain size of pure Mg reaches 2.6 μm and 1.4 μm after eight passes of ECAP at 200 °C and 150 °C [50]. Figure 7 shows the grain distribution at different temperatures. The lower the temperature, the smaller the grain size and the more uniform the microstructure. This is because the low temperature can suppress dynamic grain growth and dynamic recrystallization [49]. Due to the high temperature and long pass time of Mg alloys treated at high temperatures, recrystallization will occur and the grains will grow rapidly [51,52]. Minárik et al. showed that temperatures between 180 °C and 250 °C generally result in average grain sizes greater than 1μm for the Mg alloys [53].

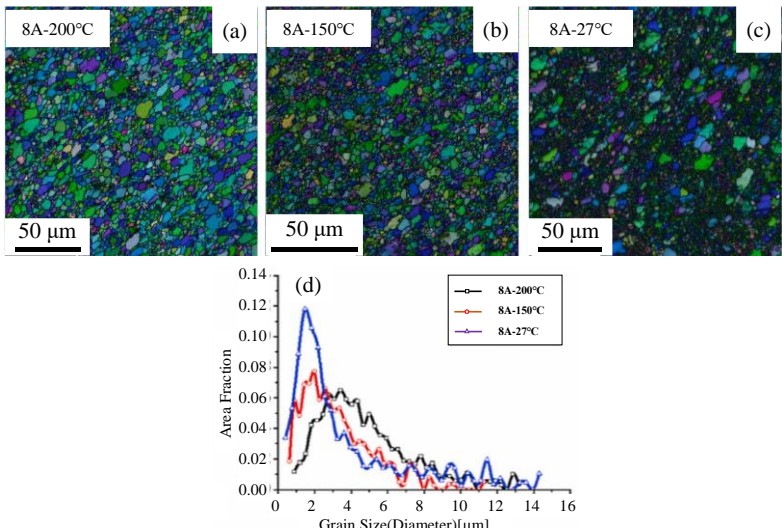

**Figure 7.** Inverse pole plot at different temperatures [50]: (**a**) 200 °C; (**b**) 150 °C; (**c**) 27 °C; (**d**) is the area fraction particle size distribution at different temperatures. Reproduced with permission from Elsevier B.V.

Consequently, grain sizes ranging from 1 to 10 µm can usually be achieved through four passes of ECAP for titanium and magnesium alloys, while two passes are required for pure aluminum. Furthermore, the Bc route has the best grain refinement effect among the four traditional routes, and low processing temperatures usually result in small grain sizes.

### 2.1.2. Grain Size from 0.1–1 µm

Further increase in the ECAP passes normally gives rise to smaller grain sizes. Hyun et al. found that the average grain size of pure Ti reached 1 µm after two passes of ECAP, and the grain size decreased to 0.4 µm and 0.3 µm after four passes and six passes at 683 K and Bc route conditions [1]. Liu et al. Obtained a more uniform equiaxed grain size in four passes using route C at room temperature (0.17 µm) [54]. On the Bc route and at temperatures around 400 °C, Zhilyaev et.al. and Fan et.al performed eight passes of ECAP on pure Ti to reduce the grain size to 0.4 µm and 0.5 µm, respectively [55,56]. Figure 8 shows the microstructure changes of pure titanium at different passes. Hajizadeh et al. performed ten passes of ECAP, and the average grain size was reduced to 183 nm, which is more significant than in previous studies [57]. For Ti and its alloys, under different conditions, the grain size can reach a sub-micron level after four passes, and the grain size decreases correspondingly with the increase of passes.

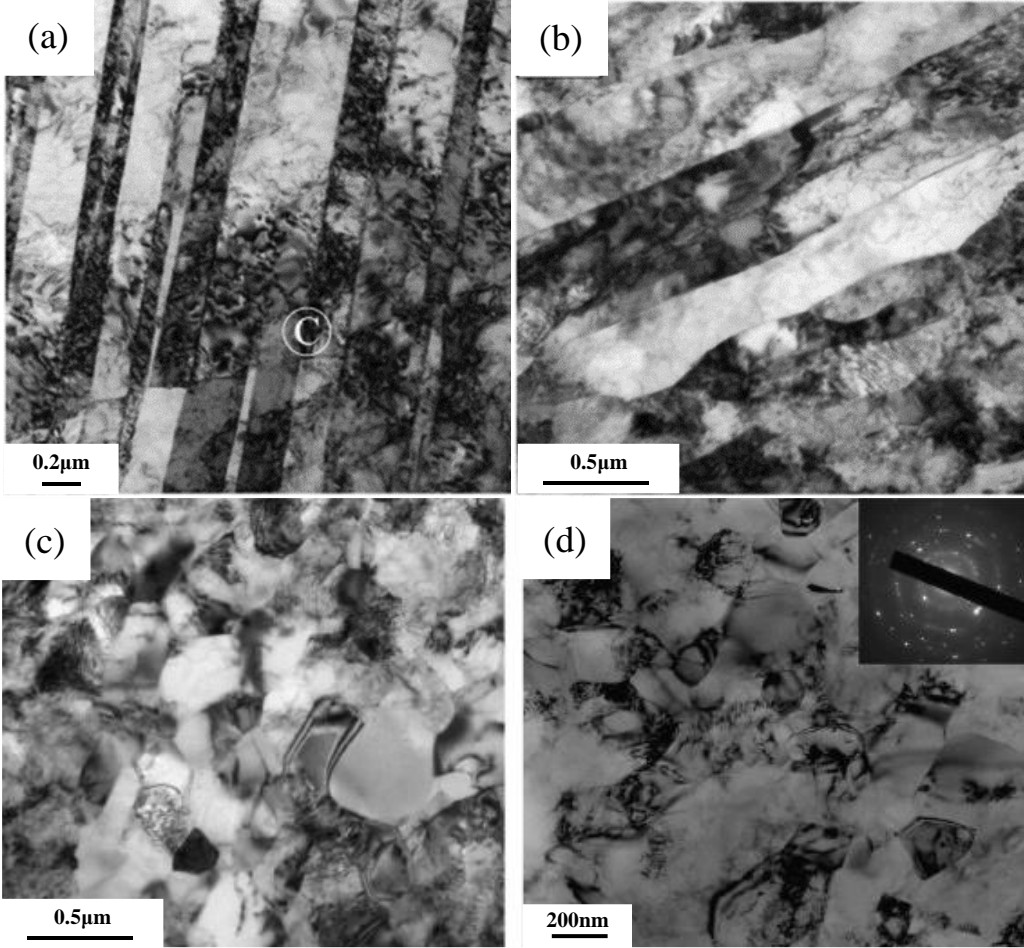

**Figure 8.** Microstructure of pure titanium under different passes [55]: (**a**) one pass; (**b**) two passes; (**c**) four passes; (**d**) eight passes. Reproduced with permission from Elsevier B.V.

Similar to titanium, the grain size of Al can be reduced to a sub-micron scale with the increasing pass. Aal et al. showed that the average grain size of pure Al was 1.7 µm after two passes of ECAP and decreased to 0.43 µm and 0.23 µm after four and ten passes at

room temperature through route A, respectively [58]. The same conditions are applied in his other articles paper, it was also found that the average grain size was reduced from 390 μm to 1.8 μm, 0.4 μm and 0.3 μm after two, four and ten passes, respectively [59]. Song et al. observed that the grain size of pure Al decreased from 200 μm to 0.5 μm after sixteen passes [60].

However, for Mg alloys, it is difficult to refine the grain size to submicron grains with the increase of the pass. Wang et al. showed that the grain size decreased from 160–180 μm to 6–8 μm after twelve passes at 380 °C for Mg alloys with long-period stacking structure [61], Jiang et al. reported that ultrafine equiaxed grains of 2.5 μm were obtained after sixteen passes via route A at 603 K [62]. Ma et al. found that the grain size of Mg alloys decreased from 80 μm to 1.5 μm after thirty-two passes at 603 K [63]. Figure 9 shows the microstructure and grain size histograms for different passes.

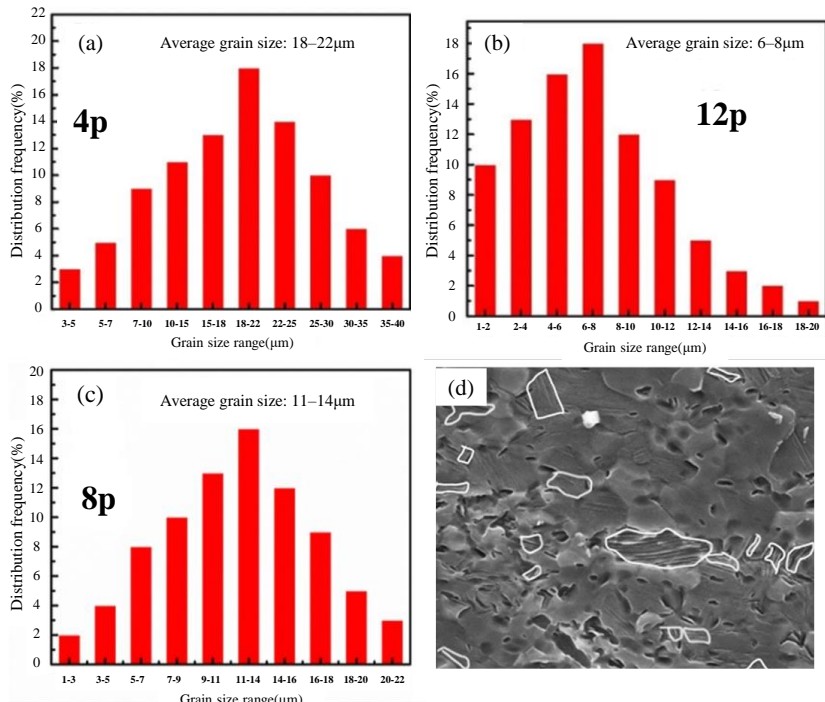

**Figure 9.** Grain size histogram [61]: (**a**) four passes; (**c**) eight passes; (**b**,**d**) is the grain size distribution map and SEM image of twelve passes. Reproduced with permission from Springer Nature.

The same tendency to transmit the effect of grain size is found in nickel and its alloys as in Al and Ti. Zhilyaev et al. reported ultrafine grains with an average size of about 0.35 μm after eight passes during ECAP pure nickel [64]. This is similar to the grain size (0.3 μm) reported by Neishi et al. for eight passes of pure Ni at room temperature [65]. The grain size was further reduced to about 0.24 μm when the passes increased to twelve passes [66]. Moreover, the grain size of Fe–Cr–Ni alloy was reduced to 0.4 μm after eight passes of ECAP processing, achieving a submicron-scale microstructure [67]. It appears that Ni and its alloys are more likely to reach submicron structures after eight passes.

The grain size of the copper and its alloys also decreases with the increase of passes. Jayakumar et al. refined the Cu-Cr-Zr alloy to 0.15~0.2 μm after eight passes of pressing, and it has high thermal stability [68]. Wongsa-Ngam et al. showed that the Cu-0.1% Zr alloy also formed a submicron structure after eight passes [69]. Khereddine et al. evaluated the grain size of Cu–Ni–Si alloy after high-pressure torsion (HPT) and ECAP treatment. The grain size after twelve passes of ECAP treatment was 0.2 μm [70]. Abib et al. found that the deformed microstructure evolved from fibrous cast grains to almost equiaxial microstructure after sixteen passes of pressing, and the high angle grain boundary fraction increased with the increase of ECAP passes [18]. The grain size of oxygen-free

pure Cu decreased to 0.6 µm after twenty-four passes of equal-pass angular pressing at room temperature, but an increase in grain size was observed with increasing strain after sixteen passes because of dynamic recovery [71]. Figure 10 shows the microstructure and orientation distribution of the Cu alloy after different passes. For copper and its alloys, the grain size can also be reduced to the submicron level after eight passes.

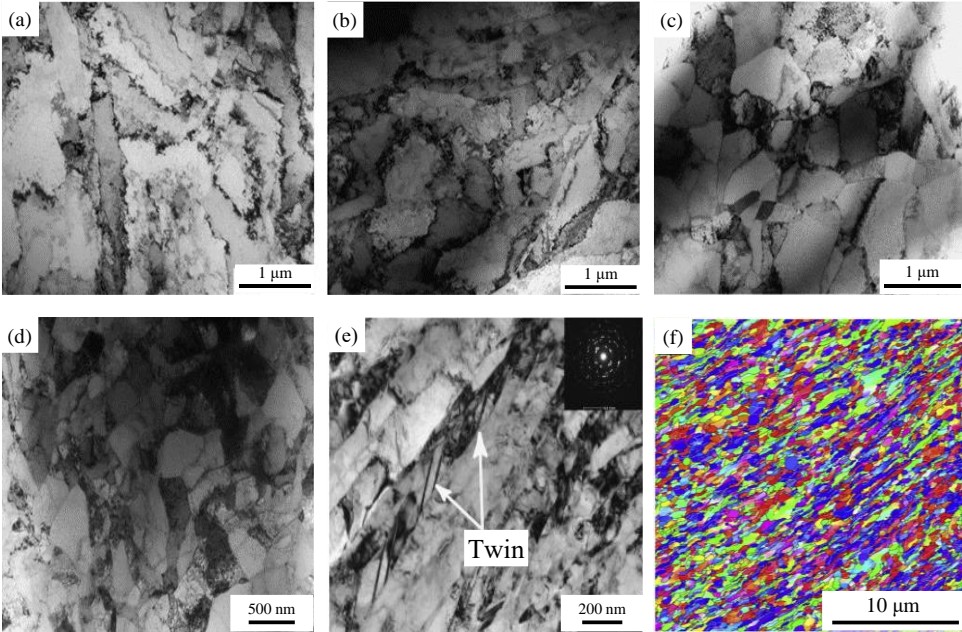

**Figure 10.** TEM and EBSD images of the microstructure of Copper Alloys after ECAP for (**a**) one pass; (**b**) two passes; (**c**) four passes; (**d**) eight passes [69]. Reproduced with permission from Elsevier B.V. (**e**) twelve passes [70]. Reproduced with permission from Elsevier B.V. (**f**) sixteen passes [18]. Reproduced with permission from Elsevier Inc.

Schafler et al. studied the evolution of Cu lattice defects under different routes. With the increase of deformation, the corresponding dislocation densities of different routes are different and show an increasing trend. The comparison shows that the Bc route has the largest dislocation density under the same strain [72]. Purcek et al. carried out eight passes of ECAP under different routes, and all three routes finally refined the Cu-Cr-Zr alloy to the submicron structure. The grain size is reduced to about 0.2–0.3 µm under three routes, but the Bc route obtained the smallest grain size and the best mechanical properties as well as good thermal stability [73]. The same conclusion was found in pure Ti [74] and pure Al [75]. Figure 11 shows the microstructures under different route treatments.

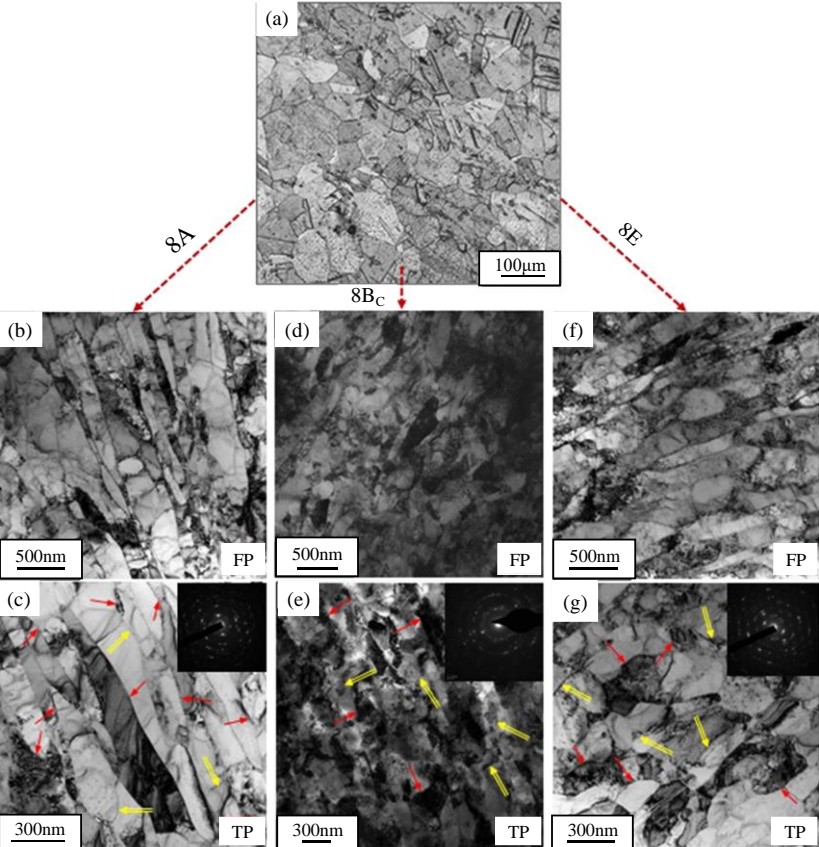

**Figure 11.** TEM images of the microstructure of Cu Alloys after eight passes of ECAP [73]: (**a**) is the pre-ECAP microstructure; (**b**,**c**) is the lateral and flow surface of the A route; (**d**,**e**) is the lateral and flow surface of the Bc route; (**f**,**g**) are the lateral and flow surface of the E route. Reproduced with permission from Elsevier B.V.

Bulutsuz et al. obtained the average grain size of pure Ti after two passes of ECAP at the composite temperature, which were 1.5 μm and 1.7 μm, respectively [76]. Attarilar et al. found that the average grain size of pure Ti reached 1.09 μm after four passes of ECAP at 400 °C [77], Ebrahimi and Attarilar measured an average grain size below 0.5 μm after four passes of ECAP at 450 °C [78]. However, Zhao et al. reported that the average grain size reached 0.25 μm after four passes of ECAP at room temperature, and even decreased to 0.2 μm after six to eight passes, resulting in a better microstructure than pure Ti treated at high temperature [79]. Similar conclusions have been obtained in Ti-Ni alloys, where lower temperatures lead to smaller grain sizes [80]. Under the low-temperature condition of liquid nitrogen treatment, the grain size of pure Ti is reduced to 0.56 μm after ECAP processing [81]. Psaruk and Lapovok studied the microstructure of pure Ti at different temperatures. As the temperature increases, more high-angle grain boundaries appear. However, the decrease in ECAP temperature leads to an increase in strength properties, indicating that the material undergoes thermally activated plastic deformation [82]. It can be seen that the temperature has a certain influence on the ECAP processing technology. Figure 12 shows the microstructure and grain size distribution of pure Ti at different temperatures.

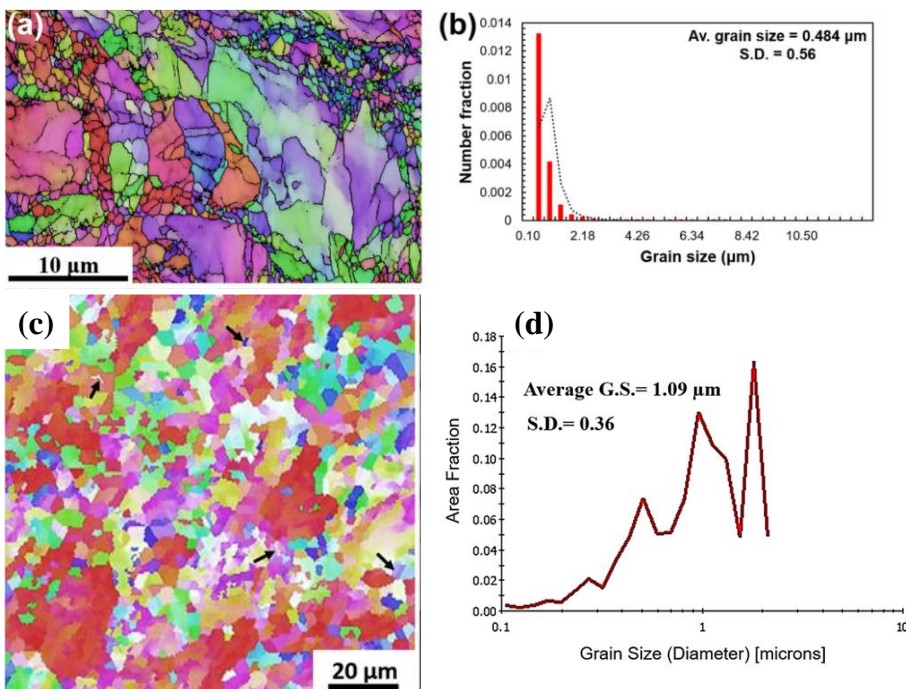

**Figure 12.** Microstructure of 4-pass pure titanium at different temperatures and grain size distribution:(**a**,**b**) at 450 °C [78]. Reproduced with permission from Springer Nature. (**c**,**d**) at 400 °C [77]. Reproduced with permission from Elsevier B.V.

Applying back pressure during ECAP can improve the microstructure. The grain size of no back pressure and after applying 100 MPa back pressure is 0.24 μm and 0.2 μm, respectively. Applying a certain back pressure to electrolytic hard-pitch pure Cu can reduce the grain size and increase the proportion of high-angle grain boundaries [83]. Similar conclusions were obtained for ECAP with back pressure applied in Al alloys [84,85] and Ti alloys [86]. The grain size of AA5083 alloy was even reduced to 0.25 μm after three passes of ECAP under 200 MPa back pressure [87]. Increasing back pressure will reduce the thermal stability of the material, but the effect of back pressure on the improvement of the mechanical properties of the material makes this defect worthless [83]. Figure 13 shows the microstructural evolution of Al alloys with and without back pressure.

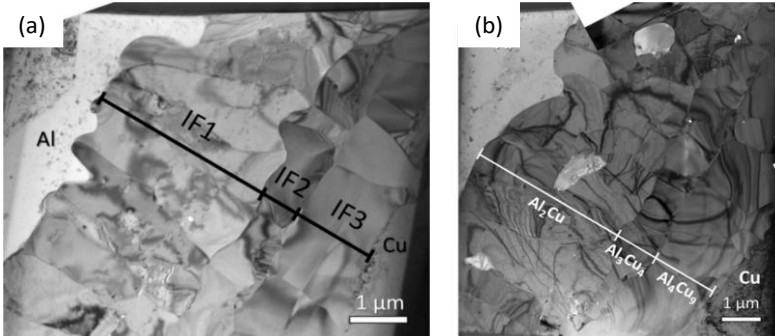

**Figure 13.** TEM images and SAED patterns of the substructure evolution after ECAP [88]: (**a**) no back pressure; (**b**) with back pressure. Reproduced with permission from Elsevier Ltd.

ECAP technology has a high strain rate during processing, which will improve production efficiency [89]. Demirtas et al. achieved maximum superplasticity at high strain rates and obtained the smallest grain size (0.2μm) of the Zn-22 Al alloy using two-step ECAP [90]. Afifi et al. refined the grain to 0.8μm and 0.3μm after one pass and four passes at different strain rates, respectively [91]. At different strain rates, it is found that the in-

crease of strength is due to grain refinement, high dislocation density and more precipitates. With the increase of passage and temperature, the strain rate sensitivity will be enhanced, and the strength after dynamic ECAP is higher than that of quasi-static flow [92,93].

Consequently, most metal materials can obtain submicron grain after four to eight passes of ECAP. At the same time, it is found that the Bc route still has the best grain refinement effect. Under the condition of not exceeding the recrystallization temperature, choosing the appropriate temperature is beneficial to grain refinement. Moreover, the increase of back pressure is beneficial to the decrease of grain size. Compared with quasi-static ECAP, dynamic ECAP will obtain higher strength and improve machining efficiency.

### 2.1.3. Grain Size below 0.1 μm

Aal et al. found in their study of Al-Cu alloys that the pass and Cu content as well as the homogenization process can affect the average grain size. For heterogeneous Cu-Al alloys, the average grain size obtained gradually decreases but is larger than 0.1 μm with increasing pass and Cu content. The copper-aluminum alloy obtained by the homogenization process, after two passes, four passes and nine passes of ECAP processing, the average grain size decreased from 261 μm before ECAP to 177 nm, 165 nm and 55 nm, respectively. Homogeneous alloys have better structure and better deformation uniformity than heterogeneous alloys. It can be seen that the increase of the homogenization treatment and the number of passes can obtain smaller grain sizes, and even obtain nano-scale grains [94,95]. Figure 14 shows the variation trends of grain size and high-angle grain boundaries under different conditions.

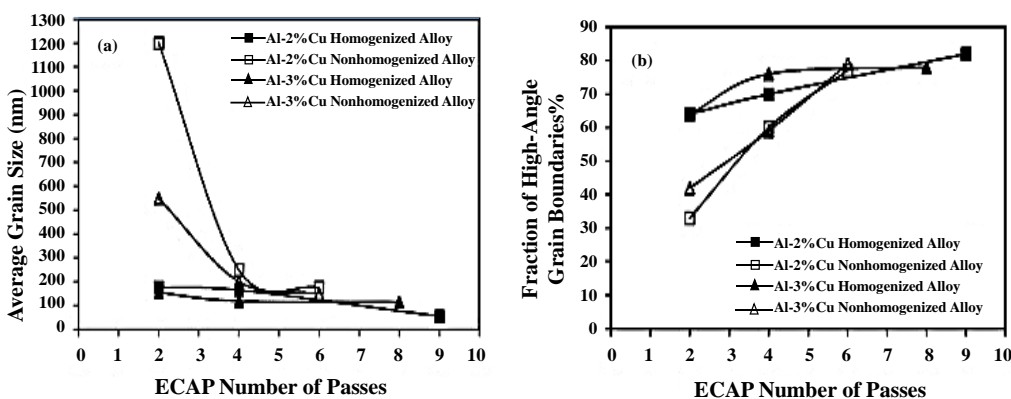

**Figure 14.** (**a**) shows the average grain size of the alloy as a function of the pass, and (**b**) shows the high angle grain boundary ratio of the alloy as a function of the pass [95]. Reproduced with permission from Elsevier B.V.

Stacking fault energy (SFE) plays a role in the fabrication of ultrafine and nanocrystallites. Lower stacking fault energies allow for the formation of finer microstructures. Qu et al. studied the microstructure of Cu-Al alloys at different stacking fault energies. The value of SFE will also be different for different alloys with Al content. They found that the grain size of Cu-5% Al alloy reached 107 nm after four passes of ECAP, and the average grain size of Cu-8% Al alloy reached 82 nm under the same conditions. Figure 15 shows the grain size distribution of Cu alloys with different Al contents after four passes. With the increase of Al content, the average grain size of the alloy decreased to the nanometer scale, and the microstructure was more uniform [96,97].

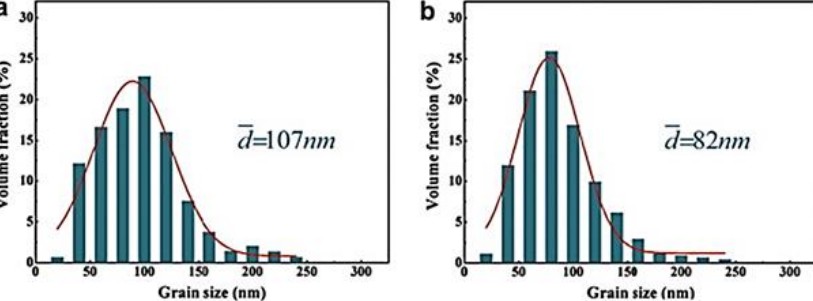

**Figure 15.** The statistic grain size of alloys after four-pass pressing measured by TEM: (**a**) Cu–5 at.% Al; (**b**) Cu–8 at.% Al [96]. Reproduced with permission from Elsevier Ltd.

Homogenizing the Al alloy will improve the machinability of the material, and a smaller grain size will be obtained after the ECAP process. For Cu alloys, the grain size can be reduced to the nanometer level by changing the stacking fault energy when other processing parameters reach the extreme. Smaller average grain size and uniform microstructure contribute to better metal properties.

It is difficult for the ECAP process to refine grains down to the nanometer level. The effect of processing factors such as pass, route and temperature to refine the grain will reach a limit, and too much change will make the grain coarser. Obtaining nanoscale grain size through the ECAP process requires not only starting from external conditions (changing processing factors, adding other processes, etc.) but also understanding the limitations of the nature of the material itself in refining grains. Table 1 summarizes the changes of grain size of different materials after ECAP process with different processing parameters.

**Table 1.** Effects of ECAP parameters on the grain size evolution.

| Material | Initial D/μm | Passes | Routes | Temperature | D after ECAP/μm | Refs |
|---|---|---|---|---|---|---|
| Ti alloy | 30 | 4 | $B_C$ | 723 K | 10 | [32] |
| Ti alloy | 15–20 | 4 | $B_C$ | 723 K | 5–10 | [33] |
| Ti alloy | 50 | 6–8 | \ | 500 °C | 0.3–0.5 | [80] |
| Pure Ti | 2000 | 4 | $B_C$ | 573 K | 2.82 | [47] |
| Pure Ti | 26 | 4 | C | Room T | 0.17 | [54] |
| Pure Ti | 40–120 | 8 | $B_C$ | 390–400 °C | < 0.5 | [55] |
| Pure Ti | 10 | 8 | $B_C$ | 450 °C | < 0.4 | [56] |
| Pure Ti | 20 | 10 | \ | 250 °C | 0.183 | [57] |
| Pure Ti | \ | 8 | $B_C$ | 400–450 °C | 0.26 | [74] |
| Pure Ti | 20 | 4 | $B_C$ | 400 °C | 1.09 | [77] |
| Pure Ti | 24.96 | 6 | $B_C$ | 450 °C | 0.29 | [78] |
| Pure Ti | 23 | 8 | $B_C$ | Room T | 0.2 | [79] |
| Pure Ti | 196 | 4 | $B_C$ | Liquid nitrogen T | 0.56 | [81] |
| Mg alloy | 20.4 | 4 | $B_C$ | 275 °C | 3.9 | [34] |
| Mg alloy | 50 | 4 | $B_C$ | 573 K | 10 | [35] |
| Mg alloy | 45 | 4 | $B_C$ | 275 °C | 1 | [36] |
| Mg alloy | 4.3 | 4 | A | 250 °C | 1.3 | [43] |
| Pure Mg | 200 | 4 | $B_C$ | 250 °C | 6–8 | [42] |
| Pure Mg | \ | 4 | $B_C$ | 225 °C | 1.896 | [45] |
| Pure Mg | 12 | 8 | A | 27 °C | 0.75 | [50] |
| AZ80 | 50.2 | 4 | $B_C$ | 390 °C | 12.82 | [46] |
| AZ31 | 75 | 8 | $B_C$ | 200 °C | 0.7 | [48] |
| AZ31 | 24 | 6 | $B_C$ | 553 K | 4.8 | [49] |
| ZK60 | 37 | 4 | $B_C$ | 250 °C | 10.9 | [51] |
| ZM21 | 5–60 | 4 | $B_C$ | 200 °C | 0.7 | [52] |

| | | | | | | |
|---|---|---|---|---|---|---|
| ZE41A | \ | 16 | A | 603 K | 2.5 | [62] |
| ZE41A | 80 | 32 | \ | 603 K | 1.5 | [63] |
| Pure Al | 1000 | 4 | $B_C$ | Room T | 1.3 | [25] |
| Pure Al | \ | 4 | $B_C$ | Room T | 1.2 | [37] |
| Pure Al | 39 | 2 | $B_C$ | Room T | 7.4 | [41] |
| Pure Al | 300 | 8 | C | \ | 2.9 | [44] |
| Pure Al | \ | 10 | A | Room T | 0.23 | [58] |
| Pure Al | 390 | 10 | A | Room T | 0.3 | [59] |
| Pure Al | 200 | 16 | \ | Room T | 0.5 | [60] |
| Al-SiCp | 55 | 2 | $B_C$ | Room T | 8 | [40] |
| AA5083 | \ | 3 | $B_C$ | Room T | 0.25 | [87] |
| Al alloy | 1.3 | 4 | $B_C$ | 423 K | 0.3 | [91] |
| Al-Cu | 261 | 9 | A | Room T | 0.055 | [95] |
| Cu alloy | 40–80 | 8 | $B_C$ | Room T | 0.15–0.2 | [68] |
| Cu alloy | 30 | 8 | $B_C$ | Room T | 0.33 | [69] |
| Cu alloy | \ | 12 | A | 423 K | 0.2 | [70] |
| Cu alloy | 24 | 24 | $B_C$ | Room T | 0.6 | [71] |
| Cu alloy | 55 | 8 | $B_A + C$ | Room T | 0.3 | [73] |
| Pure Cu | 28 | 12 | $B_C$ | Room T | 0.24 | [83] |
| Cu-Al | \ | 4 | $B_C$ | Room T | 0.082 | [96] |

*2.2. High/Low Angle Grain Boundary Evolution*

High angle grain boundaries are when the misorientation angle is greater than 15°, while low angle grain boundaries are when the misorientation angle is between 3° and 15°. It is well known that the formation of high/low angle grain boundaries contributes to grain refinement, thereby improving the mechanical properties of materials.

For the FCC metals represented by Al and Cu, grain boundary angle has a certain tendency of evolution with increasing pass times. Khelfa et al. studied the grain boundary angle change of Al alloy (AA6061) [98]. Figure 16 shows the grain boundary angle distribution of AA6061 alloy under different passes. It can be seen from the figure that with the increase in the number of passes, the proportion of high-angle grain boundaries increases, but the proportion of low angle grain boundaries decreases. Alateyah et al. believed that there is a certain relationship between dislocations and grain boundary angle in the ECAP process, i.e., the formation and development of dislocations promote the formation of high and low angle grain boundaries, and dislocation slip will be inhibited when the high/low angle grain boundary reaches a certain proportion [13]. And Reihanian et al. found the same trend of grain boundary angle in pure Al [99]. Zhang et al. gave the change curve of grain boundary angle and dislocation of Cu and its alloys with the pass as shown in Figure 17 [97]. Therefore, the evolution of the grain boundary angle can be seen more intuitively.

For HCP metals represented by magnesium and titanium, studies have shown that the changing trend of grain boundary angle in Mg alloys [100] and pure Ti [101] is similar to that of FCC metals. Li et al. found that the phase change in Mg alloys with the pass also affects the transition between high and low angle grain boundaries. The twisting and fragmentation of the internal phase of the grains not only facilitates the transformation of low-angle grain boundaries to high-angle grain boundaries but also refines the grains [102].

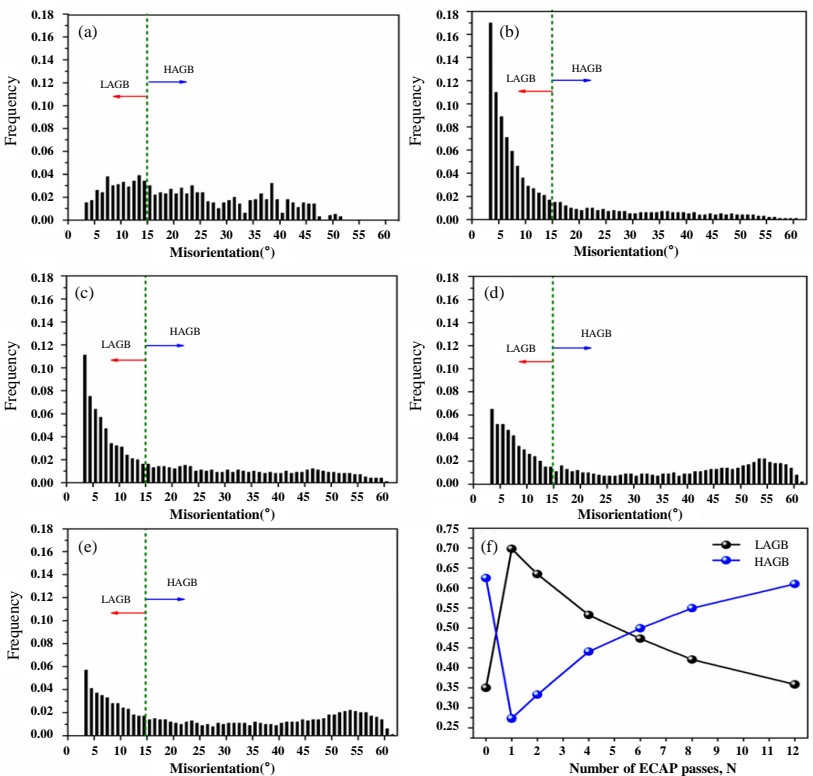

**Figure 16.** Variation of angular grain boundary of aluminum alloy after different passes [98]: (**a**) zero  pass; (**b**) one pass; (**c**) four passes; (**d**) eight passes; (**e**) twelve  passes; (**f**) is the angular grain boundary variation curve of different passes. Reproduced with permission from Springer Nature.

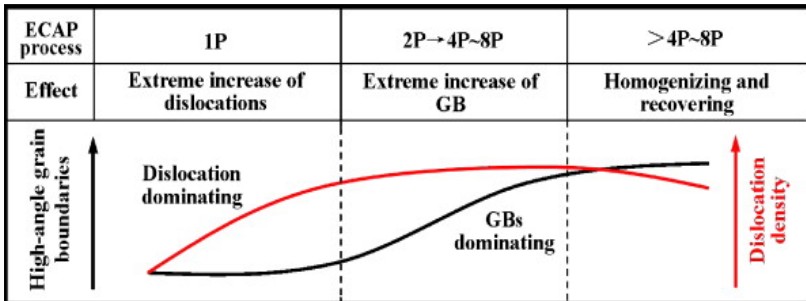

**Figure 17.** Variation trends of angular grain boundaries and dislocations in copper and its alloys [97]. Reproduced with permission from Elsevier B.V.

Low-angle grain boundaries have been reported to play an important role in the strengthening of alloys [98]. Luo et al. quantitatively studied the strength contribution of high and low-angle grain boundaries to pure Ti and found that the Hall-Petch law largely follows the strength contribution of low-angle grain boundaries [103]. However, Balasubramanian and Langdon believed that high-angle grain boundaries play a role in ameliorating the microstructure, and low-angle grain boundaries have little effect on the strengthening of materials because low-angle grain boundaries are more likely to produce slip and have a lower barrier effect [104]. Furthermore, Qarni et al. believe that existing high-angle grain boundaries provide nucleation sites, while low-angle grain boundaries transform around them. And they think that grain refinement starts from the outside and gradually spreads to the inside [105].

In different alloys, after a certain strain deformation, the high-angle grain boundary and the low-angle grain boundary are no longer transformed, and an equilibrium state is

reached [106]. Due to dynamic recrystallization at the grain boundaries, a stable micro-structure will be formed [107]. The grain refinement also reaches the extreme when the deformation of the alloy reaches saturation [108].

### 2.3. Phase

#### 2.3.1. Titanium Alloys

For ($\alpha$ + $\beta$) two-phase titanium alloys, Suresh et al. found that the evolution of the two phases is different for different passes [109]. By observing the texture evolution of $\alpha$ phase for the first four passes, they think it is the result of dynamic recrystallization. After four passes, the $\alpha$-phase texture weakens and the $\beta$-phase texture appears, which is due to the local lattice rotation of the $\beta$-phase around the $\alpha$-phase. After eight passes, the $\beta$ phase formed an ideal texture structure, indicating that the delayed dynamic recrystallization of the $\beta$ phase occurred [109]. Figure 18 shows the evolution of the two phases after different passes. Kocich et al. also showed that the phase evolution would be affected by the increase of pass number and the decrease of deformation temperature in special titanium alloys [110]. In the study of the $\alpha$ phase, it was found that the high dislocation density and a large number of non-equilibrium grain boundaries/sub-grain boundaries generated during the ECAP process, which provide a good nucleation site for $\alpha$-phase precipitation, and the ultrafine equiaxed $\alpha$ grains were formed in the $\beta$ matrix. Moreover, vacancies and non-equilibrium grain boundaries in the microstructure enhance atomic diffusivity and accelerate the formation of the $\alpha$-phase [111]. Dyakonov et al. argued that the evolution of the primary $\alpha$ phase was driven by continuous dynamic recrystallization [112]. Polyakova et al. also found that the fragmentation of the primary $\alpha$-phase grains was achieved due to the slip and accumulation of dislocations [113].

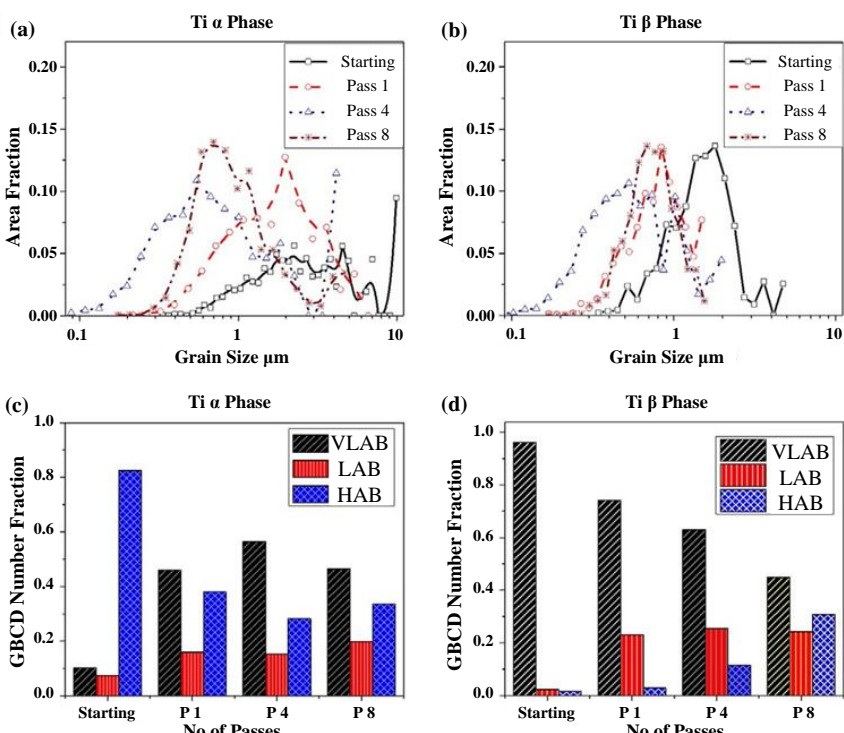

**Figure 18.** Grain distribution and angular grain boundary distribution of $\alpha$ and $\beta$ phases under different passes [109]: (**a**,**c**) is the $\alpha$ phase and (**b**,**d**) is the $\beta$ phase. Reproduced with permission from Elsevier Inc.

The slender primary $\alpha$ phase was formed in hot extruded Ti alloy after ECAP treatment, and the primary $\alpha$ slender grains play an important role in the failure behavior of

Ti alloy. Semenova et al. found that the lowest fracture toughness was obtained in the sample with elongated grains parallel to the loading direction [114]. Bartha et al. showed that the formation of the alpha phase is accelerated in regions with a higher concentration of lattice defects. The equiaxed $\alpha$-phase is mainly formed in the area with a dense grain boundary network and high dislocation density, and the flaky $\alpha$-phase particles precipitate along the coarse β-grain boundary, which inhibits the precipitation of the $\alpha$-phase inside the β-grain [115].

Sun and Xie reported that the β phase transformed into an acicular martensite $\alpha'$ phase after ECAP processing, which was helpful for further grain refinement in other phase regions [116]. Li et al. also found that after ECAP processing, the preferred orientations vary. Figure 19 shows the XRD images of the phases under different conditions. A shift from the preferred orientation of the β(200) peak to a combination of the β(200) peak and the β(211) peak as the preferred orientation. The different intensities of the β peaks in different regions are considered to be caused by the changes in the temperature field and stress field during the ECAP process. They also showed that the deformation mechanism of hot-extruded Ti alloys includes stress-induced transformation of $\alpha''$ phase to β phase, and it is believed that the deformation mechanism of Ti alloys depends to a large extent on the grain size of the β phase [117].

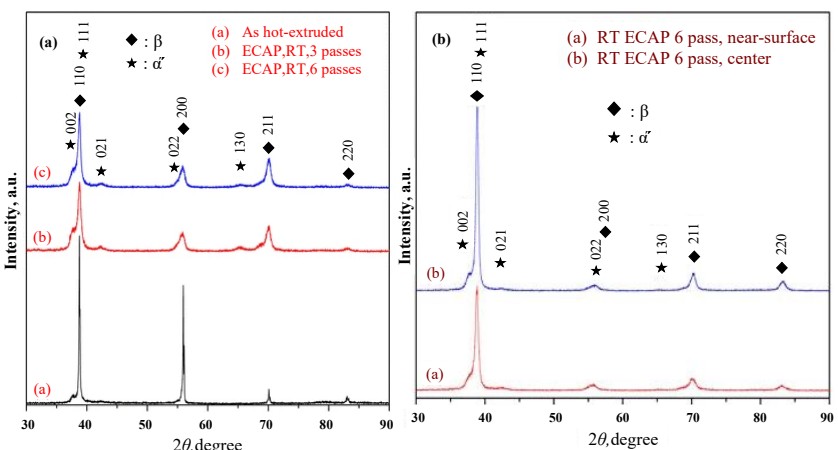

**Figure 19.** XRD patterns of different phases [117]: (**a**) Different processing passes; (**b**) different regions. Reproduced with permission from Springer Nature.

### 2.3.2. Mg Alloys

Phase evolution affects the microstructure and properties of metallic materials during ECAP. Liu et al. found that ECAP could refine the grain size of each phase but could not change the dispersion of the phases. And they found that the refining rate of the β phase is higher than that of the $\alpha$ phase in Mg alloys [118]. Both phases are more uniform as the number of passes increases [119]. Moreover, Xu et al. also found that the formation of the β phase can inhibit grain growth [120]. Figure 20 shows the microstructure and (0002) pole figures of Mg alloys at different passes. For Mg alloys containing second-phase particles, ECAP can promote the refinement of the matrix $\alpha$-phase, but after too many passes, the $\alpha$-Mg grains will grow and form a block, which is caused by the decline of the mechanical properties of Mg alloys due to particle dissolution [121]. However, Eyram Klu et al. demonstrated that the strong basal texture formed by the $\alpha$ phase of the Mg matrix can improve the strength of the material [122]. Mostaed et al. studied the grain size of the second phase in Mg alloys after ECAP. They found that the reduction in the grain size of the second phase also made the material microstructure more uniform [123]. Yang et al. show that the corrosion properties of Mg alloys are not only related to grain size but also the morphology and distribution of the β phase [124].

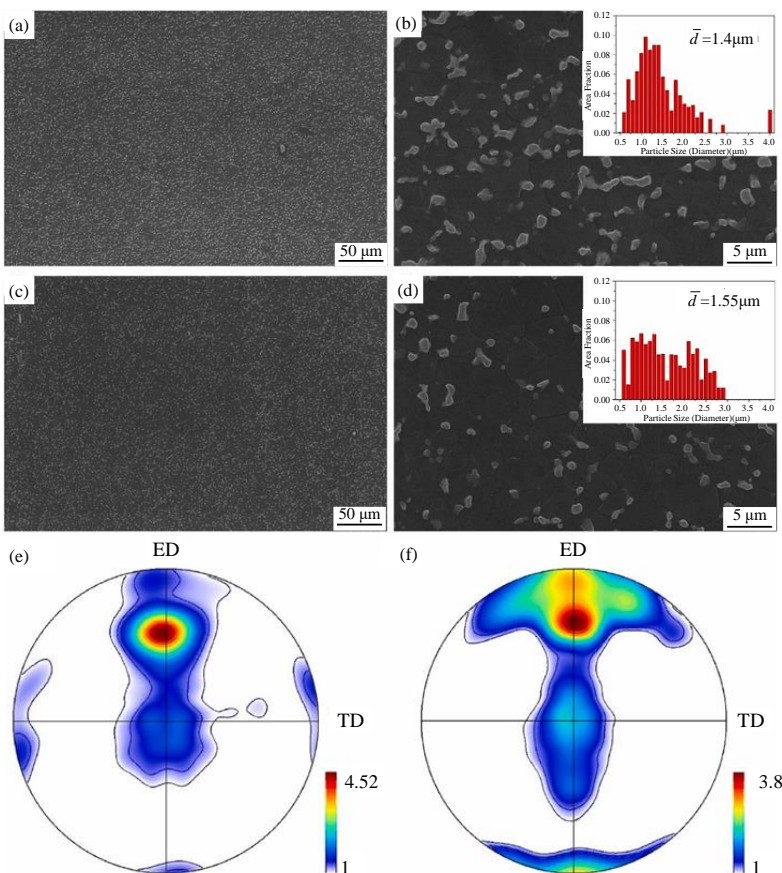

**Figure 20.** SEM images after different passes of ECAP and (0002) pole figures [120]: (**a**,**b**,**e**) four passes; (**c**,**d**,**f**) twelve   passes. Reproduced with permission from Elsevier B.V.

In addition, Furui et al. found that temperature also affects phase evolution [125]. Several phases in Mg alloys continue to grow with increasing temperatures at different temperatures. Figure 21 shows the evolution of the phases at different temperatures. As the temperature increases, the particles of different phases will grow to different degrees. Grain growth is also different at different temperature stages. When the processing temperature is at the lowest conditions, the material will obtain the most uniform and fine microstructure [126].

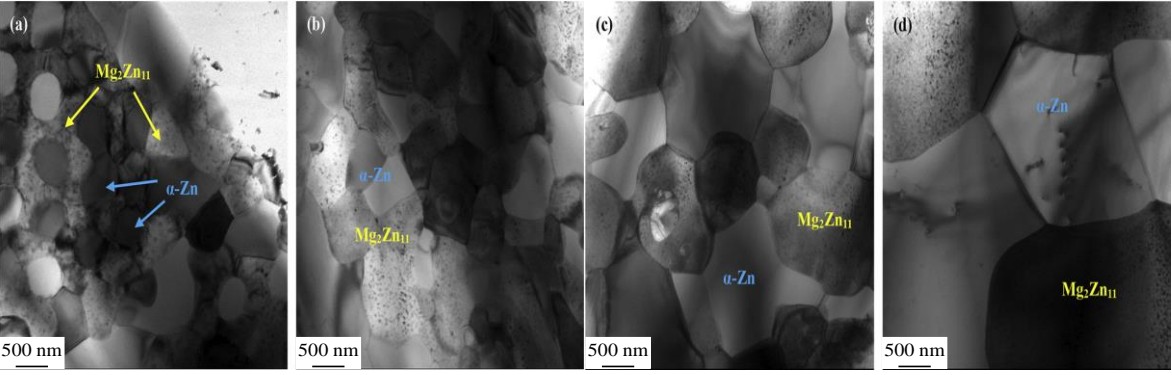

**Figure 21.** TEM images at different temperatures [126]: (**a**) 150 °C, 4 passes; (**b**) 150 °C, 12 passes; (**c**) 200 °C, 12 passes; (**d**) 250 °C, 12 passes. Reproduced with permission from Elsevier B.V.

The combination of ECAP with other treatments can affect the phase evolution and thus change the mechanical properties of magnesium alloys. Mg alloys undergo aging treatment after ECAP. Although the grain size will grow slightly and reduce the material

properties, the precipitation hardening caused by the precipitation particles will compensate for the loss and further increase the strength [127]. Yan et al. produced a homogeneous near-single-phase microstructure during the deformation process by ECAP and hot stretching techniques, thereby obtaining Mg alloys with better mechanical properties [3]. By adding the Zr element in the ECAP process, the Mg alloy can produce smaller second-phase particles, inhibit grain growth, reduce the final grain size, and improve the properties of the material [128].

### 2.3.3. Al Alloys

The formation and transformation of η and η′ phases in Al alloys can affect the properties of the material. It is found that the η phase is refined with increasing density. The η′ phase is precipitated after one pass, and many nano-scale precipitates are formed with the increase of the pass to improve the performance of the material [129,130]. And after ECAP, the size of the η′ phase in the matrix is smaller and the distribution is more uniform [131]. Temperature also affects the precipitation of the η phase and η′ phase [132]. Shaeri et al. reported that when the temperature was raised from room temperature to 120 °C, small η′ particles were formed to enhance the mechanical properties of the material. Between 120 °C and 180 °C, the mechanical properties of Al alloys decrease with the increase in temperature, which is related to the increase in the size of grains and precipitates and the transformation of the η′ phase into the η phase [133]. Xu et al. found that at 200 °C the η′ phase would partially dissolve, resulting in a reduction in the strength of the alloy, which could be recovered by aging treatment [134]. Figure 22 shows the XRD patterns of Al alloys at different temperatures.

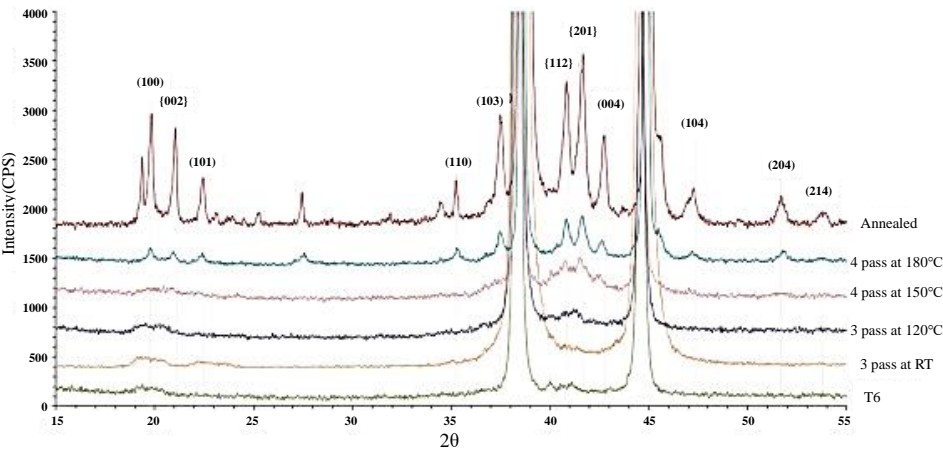

**Figure 22.** XRD patterns after ECAP treatment at different temperatures [133]. Reproduced with permission from Elsevier B.V.

Different heat treatments after ECAP also affect the phase change and alter the properties of the alloy. Wang et al. An annealing treatment after one pass of ECAP increases the hardness of the material due to the generation of very fine η′ phase particles. However, after eight passes of ECAP, the hardness of the alloy decreases due to recrystallization and the transformation of the η′ phase into the η phase. And they found that annealing at higher temperatures did not further improve the alloy properties [135]. Afifi et al. also reported that with the increase of the pass, more fine precipitates will be generated, and the increase of the annealing temperature after ECAP does not further refine the grains, but because the increase of the heat treatment temperature will cause grain coarsening [136]. Figure 23 shows the microstructures annealed at different temperatures after eight passes of ECAP. Combining the ECAP process and heat treatment optimization will produce metal materials with better properties.

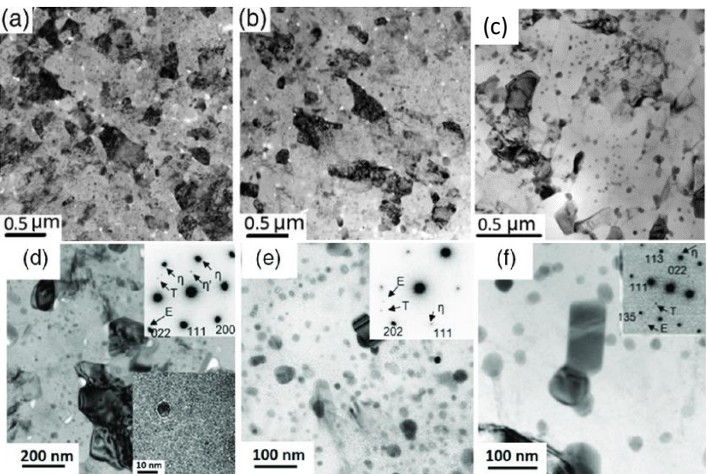

**Figure 23.** TEM images of annealing at different temperatures after 8 passes of ECAP [135]: (**a**,**d**) 393 K; (**b**,**e**) 423 K; (**c**,**f**) 473 K. Reproduced with permission from John Wiley & Sons, Inc.

## 3. Deformation Mechanism

In metallic materials, dislocations and twinning are the main deformation mechanisms [137]. At different temperatures, the deformation mechanism remains the same, which is dominated by slip [50]. Figure 24 shows the slip system for HCP metal, FCC metal and BCC metal. Chen et al. found that when the temperature is between 250 °C and 300 °C, the deformation of the alloy is controlled by grain boundary diffusion; when the temperature is between 300 °C and 350 °C, the deformation is controlled by lattice diffusion. When the temperature is further increased, dislocation creep, which requires larger activation energies, plays a role in the deformation. During the deformation process of the alloy, not all grains have a good sliding direction, and only the grains with the most favorable orientation will undergo grain boundary sliding [138]. Chen et al. studied the twinning behavior at different temperatures, as shown in Figure 25, and found that there is a certain relationship between the grain orientation and the evolution of twinning. Under the condition of variable temperature, the formation of twins causes lattice rotation and changes in grain orientation. In addition, the twin formation has different tendencies in grains with different orientations during deformation. The formation of twins changes the grain orientation, which affects the evolution of twins [27]. In the ECAP process at low temperatures, twins are the prevalent microstructure and the deformation mechanism is supposed to be induced by the twining instead of dislocation slipping. With the increase of temperature, both twinning and dislocation slipping contribute to the material deformation behavior. Moreover, the fracture of twins after increasing the temperature is conducive to forming ultrafine grains [27].

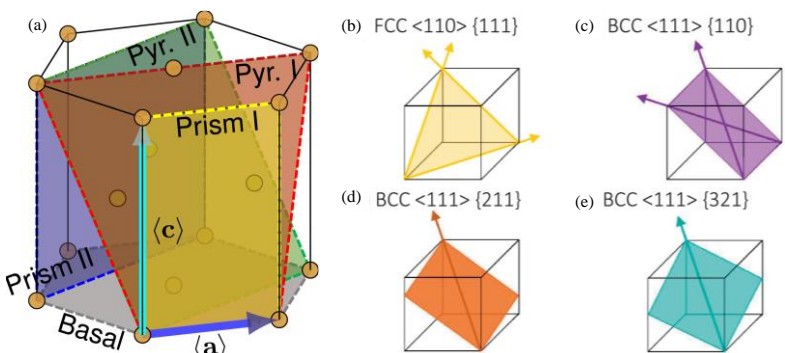

**Figure 24.** slip system: (**a**) HCP metals [139]. Reproduced with permission from Elsevier Ltd. (**b**) FCC slip systems; (**c**–**e**) BCC slip systems [140]. Reproduced with permission from Elsevier Ltd.

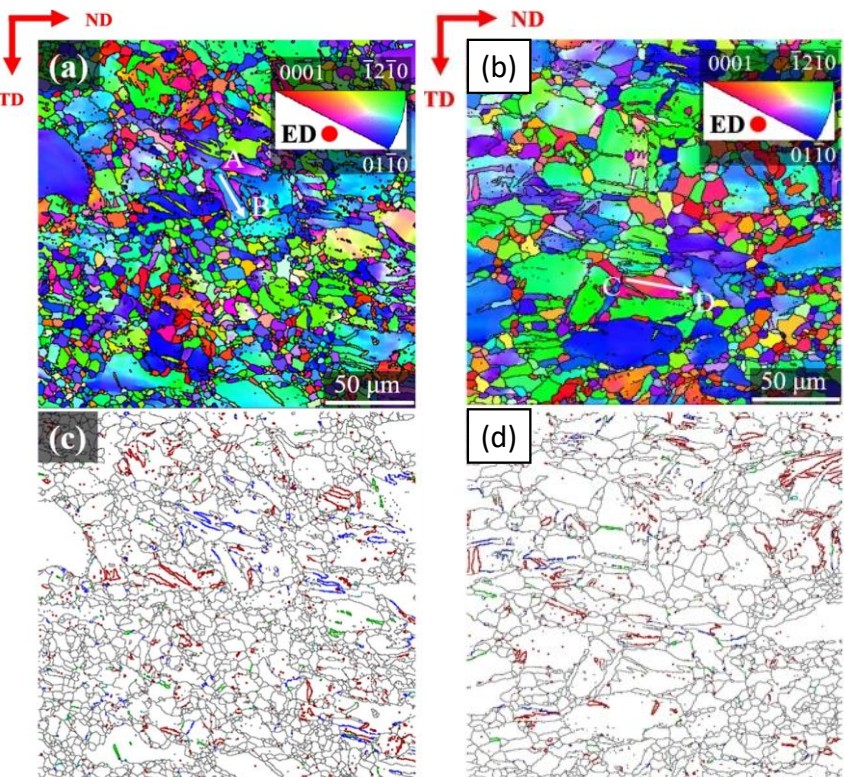

**Figure 25.** EBSD map and twin distribution map at different temperatures [27]: (**a**,**c**) 250 °C; (**b**,**d**) 300 °C. Reproduced with permission from Springer Nature.

It is found that the c/a ratio in HCP metal affects the critical shear stress (CRSS) between substrate slip and non-substrate slip, and thus affects the deformation mechanism of the material [141]. For example, the c/a of Mg is 1.624, which leads to the lower CRSS of the base plane and is conducive to the slip of the base plane. However, due to the low c/a ratio of Ti, the prism CRSS is lower, which is conducive to the prism slip [142]. Minarik also found that different c/a ratios lead to different texture evolution during ECAP processing due to the activation of different slip systems [143]. Bednarczyk et al. showed that an increase in dislocation density would also increase the CRSS of slip systems in HCP metals. The decrease of grain size will cause the deformation mechanism to change from non-basal slip to basal slip [144].

The effect of the ECAP route on the microstructure is due to the differences in the shear paths of the different routes concerning the grain extension texture plane and the direction of the crystal structure [19]. Roodposhti et al. investigated the effect of different machining routes on the ECAP processing of commercially pure titanium. They showed that the processing routes directly affect the microstructure of the material [17]. Different routes activate different slip regimes, resulting in different microstructures. Figure 26 shows the clipping paths generated by different routes. Among them, X, Y, and Z represent three different orthogonal planes, and the numbers represent the shearing directions of different passes. From the figure, we can see that the strain of route C recovers after the even number of passes, and also the third and fourth slips of route B$_C$ offset the first two slips, so route C and route B$_C$ are called redundant strain channels. The non-redundant strain passages Route A and Route B$_A$ have separate shear plane intersections, and additional shear strain accumulates with each pass. The shear directions of the C route are all switched in the orthogonal direction, resulting in equiaxed grains. Other routes deform differently in different directions because of additional shear strain and crystal anisotropy. This also explains the reason why the B$_C$ route obtains a more uniform microstructure in the experiment [14,17,19]. However, Ravisankar and Park found that route A was more effective in grain refinement for BCC alloys. This is related to the redundant strain in the

machining process. The crystal structure, slip system and fault energy can affect the route efficiency. For example, Ag, which is also an FCC metal, is more prone to twinning deformation, while Cu and Al are prone to slip deformation because of the low fault energy of Ag. HCP metals prefer c/a-based twinning due to the lack of slip systems [20].

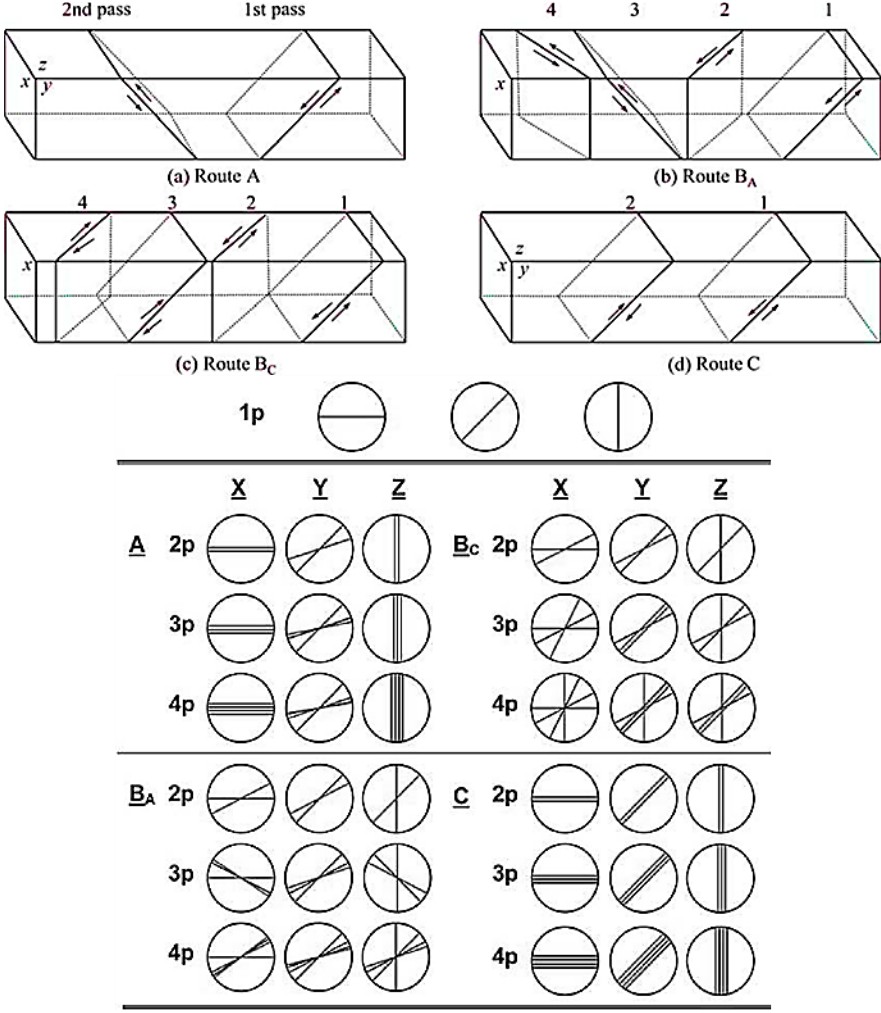

**Figure 26.** Slip system for different routes [14,20]:(**a**) Route A; (**b**) Route B$_A$; (**c**) Route B$_C$; (**d**) Route C. Reproduced with permission from Springer Nature. Reproduced with permission from Elsevier Ltd.

In the initial stage of compression deformation, twinning dominates the deformation mechanism due to the larger grain size, and as the strain increases, the grain size is refined and the twinning decreases [145]. Moreover, Xiang et al. also showed that the formation of twins can improve the deformability of HCP alloys by increasing the slip system [146]. Figure 27 shows the microstructure of the material after different passes.

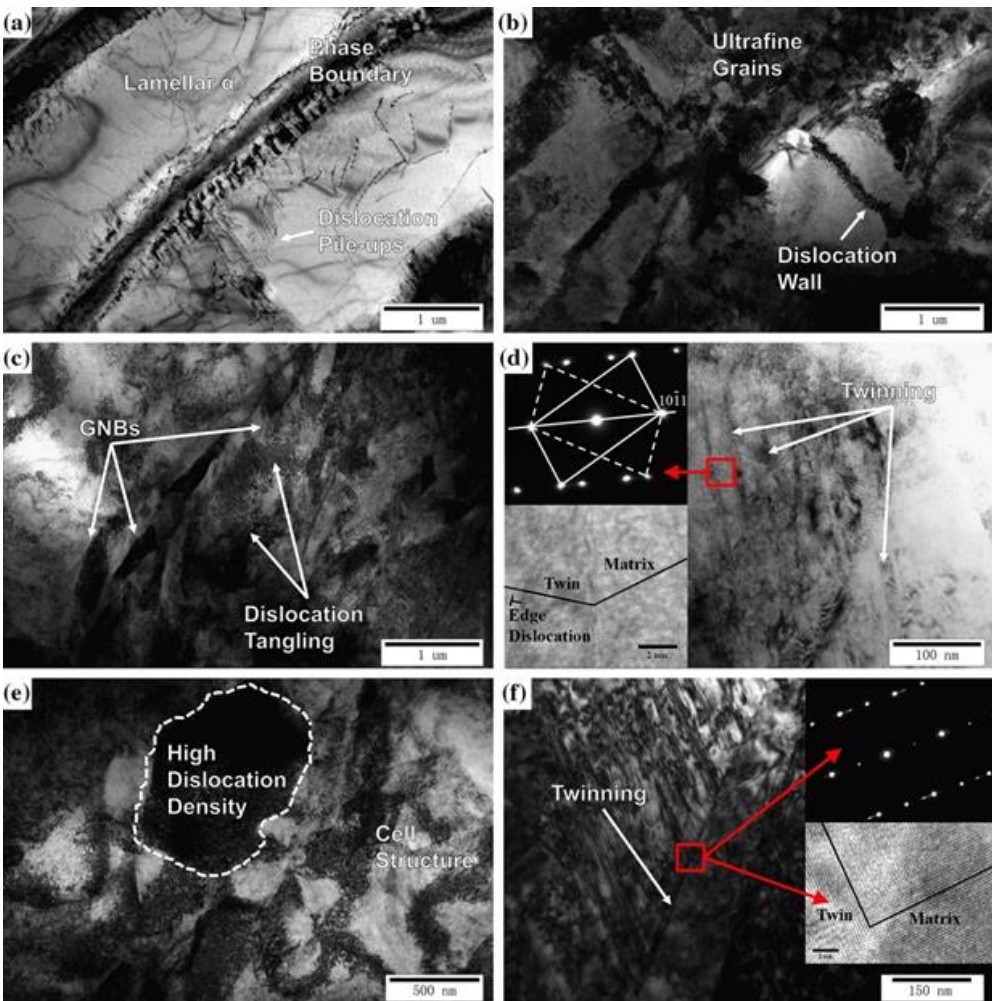

**Figure 27.** TEM images after different passes [146]: (**a**) as-received; (**b**) one pass; (**c**,**d**) two passes; (**e**,**f**) three passes. Reproduced with permission from Springer Nature.

Li et al. reported that fine grains (3 μm) can suppress the formation of twins and that the twins in the coarse grains are mostly tensile twinning, and the tensile twinning is less than the twinning and compression twinning during deformation. With grain refinement, the deformation mechanism changes from tensile twinning and <a> dislocation slip to <a> dislocation and <c + a> dislocation slip. Figure 28 shows the distribution of different dislocations [147]. Zhu and Langdon also showed that the deformation mechanism under low stress and strain is mainly grain boundary sliding and diffusion creep, while the deformation mechanism under medium and high stress and strain is mainly the slip and climbing of intragranular dislocations. When the grain size is reduced to the nanometer scale, the deformation of the material depends on its grain size. When the grain size is between 50 and 100 nm, the dislocation emission and annihilation at the grain boundary are the main deformation mechanisms, which can also explain the creep mechanism of ultrafine-grained titanium. Between 10 and 50 nm, partial dislocations and deformation twinning are the main deformation mechanisms. For grains smaller than 10 nm, grain boundary slip is still the main deformation mechanism [148,149]. Gautam and Biswas found that with decreasing grain size, the compressive mechanical behavior changed from S-shaped to parabolic, further demonstrating that the deformation mechanism changed from twinning to slipping with decreasing grain size [150].

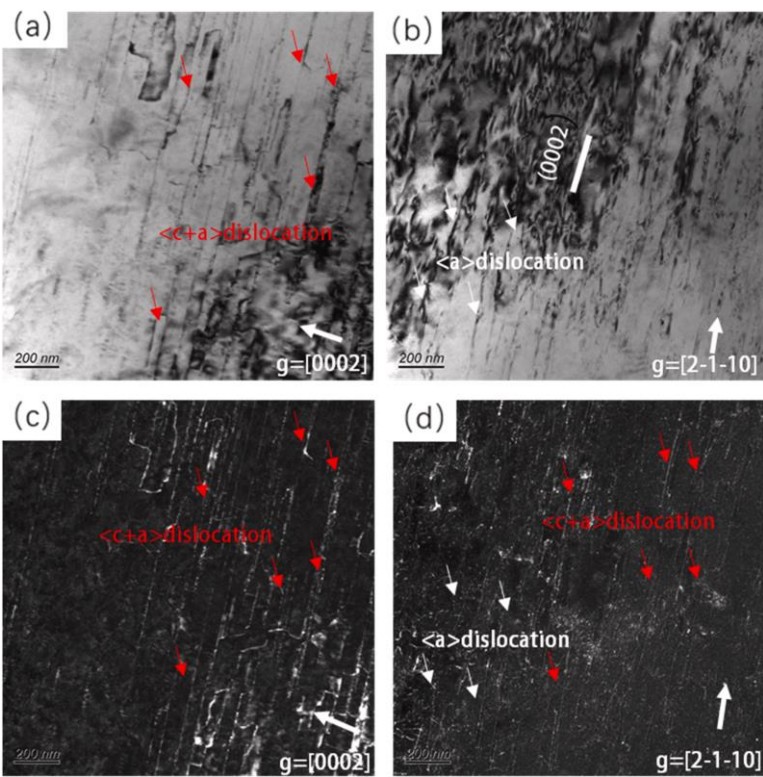

**Figure 28.** Distribution of dislocations in bright and dark fields [147]: (**a**,**b**) bright fields;(**c**,**d**) dark fields. Reproduced with permission from Elsevier B.V.

## 4. Conclusions and Perspectives

(1) Starting from the grain size, this paper discusses the processing parameters required to achieve submicron and nanoscale particles for different metal materials. The grain size required for different metal materials is obtained by analyzing the influencing factors such as processing route, pass, and temperature. The research on processing temperature and processing speed is still not comprehensive enough, and no quantitative and continuous trend of influence has been obtained. However, performing the ECAP process at a relatively low temperature is beneficial to obtain finer grains, because the low temperature can inhibit the dynamic recrystallization and grain growth of metal materials.

(2) The grain refinement of the ECAP process is via plastic deformation of the workpiece, during which grain boundaries also change simultaneously. Even though some efforts have been devoted to investigating the evolution of the high/low angle grain boundaries, the shear deformation and its interaction during grain refinement are still under discussion due to the evolution of grain boundaries in the material. Some advanced characterization techniques such as in-situ TEM and transmission Kikuchi diffraction (TKD) can provide more detail information, which are suitable candidates for investigating the grain boundary evolution in future studies.

(3) The quantitative relationship between the effective strain and the pass, channel angle, and curvature angle during ECAP processing is discussed based on previous research under ideal conditions. However, the friction in the actual machining process is not negligible, and the influence of the friction between the sample part and the mold on the ECAP process still needs to be further studied.

(4) During thermal deformation, subgrain boundaries transform from low-angle grain boundaries to high-angle grain boundaries by absorbing dislocations, resulting in grain refinement. Therefore, the dynamic recrystallization that occurs during the ECAP process is considered to be one of the main mechanisms of grain refinement. Existing studies have been able to prove that the evolution of dislocations, grain

boundary angle, phase and other factors will reduce the grain size, but no clear refinement mechanism has been given. Therefore, there is no unified consensus on the mechanism of grain refinement in the ECAP process, which still needs further exploration.

(5) In the study of ECAP, the deformation of metallic materials is controlled by twinning and slip. In general, twinning is the dominant deformation mechanism in the low-strain stage. As the strain increases, dislocation slip gradually becomes the dominant deformation mechanism of metallic materials. Under different conditions (temperature, route, pass), severe plastic deformation usually causes different changes in the dominant deformation mechanism of metal materials. Grain refinement further inhibits the formation of twins. When the grains are reduced to a certain extent, the deformation mechanism of metallic materials will change from twinning to dislocation slip.

**Author Contributions:** L.C.: Conceptualization, Writing—original draft preparation. S.S.: Methodology, Software. H.W.: Investigation, Resources. G.Z.: Investigation, Supervision. Z.Z.: Supervision, Writing—review and editing, Funding acquisition. C.Z.: Data Curation, Funding acquisition. All authors have read and agreed to the published version of the manuscript.

**Funding:** This research was funded by the Research Committee of the Shenzhen University and Shenzhen Natural Science Foundation University Stable Support Project (Grant No. 20200826160002001) as well as the National Natural Science Foundation of China (Grant No. 62003216).

**Data Availability Statement:** Not applicable.

**Conflicts of Interest:** No conflict of interest exists in this submitted manuscript, and the manuscript is approved by all authors for publication. I would like to declare on behalf of my co-authors that the work described was original research that has not been published previously, and not under consideration for publication elsewhere, in whole, or in part.

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
