# Peer review of "Recent Advances in the Equal Channel Angular Pressing of Metallic Materials"

_processes, doi:10.3390/pr10112181_

Round 1

Reviewer 1 Report

In the manuscript, the authors summarize the recent advances in equal channel angular pressing of metallic materials.

The first part of the introduction is devoted to basic explanation of the effect of grain refinement on properties of materials, while the second part focuses on processing methods themselves. In this section, the reviewer suggests mentioning the existence of novel ECAP-related SPD methods, such as twist channel angular pressing, and twist channel multi angular pressing (e.g. doi: 10.1016/j.matlet.2016.05.163, and doi: 10.1088/1757-899X/63/1/012006). The last part of the introduction is devoted to general description of the effects of ECAP itself on materials.

Section 2 focuses on detailed characterization of structure development induced by ECAP, with the special focus on Al, Mg, and Ti alloys.

p. 4 l. 22 – it is probably better to change the statement “pure Al” to “commercially pure Al”, or please state the type of alloy/material.

p. 5 – the authors are right stating that the selected deformation route affects the grain size and properties – similar conclusions were drawn by Kocich et al. (doi: 10.1016/j.matchar.2016.07.020) who, compared commercially pure Al subjected to two passes of ECAP routes A and Bc, and ECAP with implemented twist. The review could benefit from this reference.

Section 2.3.1. – in this section the authors describe the effects of ECAP on the relationship of alpha and beta phases within Ti-based alloys. The authors could also add the information that the effect of the ECAP method on phase modifications is also on special Ti-based alloys, such as the shape memory alloys (e.g. doi: 10.1016/j.jallcom.2010.12.003).

Section 3 characterizing the deformation mechanisms of metallic materials during ECAP is thoroughly written and well understandable.

Reviewer 2 Report

Very good article. As is known, after ECAP, grinding takes place mainly on the surface of the metal. Structural studies give results from the surface and X-ray phase analysis from the bulk of the material. It would be interesting to carry out structural studies from the volume of the test sample

Reviewer 3 Report

The manuscript contains a review of the results of studies of the evolution of the grain structure of metals, alloys and metal matrix composites during equal channel angular pressing (ECAP). The manuscript contains useful information on the possibility of obtaining submicron and Nano scale grain structures in metals as a result of ECAP. The review of modern studies of structural changes in metals after ECAP is of interest for a wide group of scientist, post graduate students, and students performing scientific investigations on modification of metal grain structure by a severe plastic deformation method.

The manuscript needs additions and corrections in the presentation of the results.

1) In the presented review there is practically no information about the rate of pressing of samples through the channel. The results of dynamic ECAP were not included in the review. The results of high-speed angular pressing are of interest due to their high efficiency. The degree of grain size reduction in aluminum alloys and copper which requires more than 5 passes at quasi-static ECAP is achieved in one pass of dynamic pressing.

2) Figure 1 (a) and (b) are not interpreted correctly. Figure 1(b) shows the ECAP routing schemes for square, not round, channels. Only for canals with square cross-section it is possible to realize Bc routes, because in this case it is possible to specify exactly the specified angles of axial rotation of 90 ◦ samples.

3) It should be noted that the differences of pressing technologies in orthogonal mating channels and channels mating at angles other than 90 ◦ were not reflected in the review.

4) In the analysis of the results of studies of the effect of multi pass conditions of  ECAP on the evolution of the microstructure in the subsection 2 (Lines 101-105, 113- 119, 126-131 etc.), important parameters of  ECAP processes - pressing temperature and pressing rate and the routes of pressing - were not specified.

5) It should be discussed  how the type of crystal lattice of the alloy, which determines the belonging of the material to certain isomechanical groups of materials, affects the prevailing physical mechanisms of plastic deformation and the patterns  formation of grain size distribution.

Subsection 2.1.2 does not reflect the differences in the evolution of grain structures in HPC alloys Ti and Mg with different lattice parameter ratios c/a, and Al, Fe, and Cu alloys with HCC lattice.

6) It is necessary to check the text for grammatical correctness. There are grammatical errors in the text and missing references to figures, for example:

6.1) Line 93 « blow 0.1 μm»;

6.2) Line 222 «Fig. xx».

Reviewer 4 Report

The novelty of this paper has not been explicitly given in the introduction. Please emphasize its main novelty compared with existing reviews of ECAP.

Fig. 3 needs more explanation, what is the initial grain size?

Fig. 4 needs more explanation, Figure 4 shows an image of the microstructural evolution of pure Al. Then what?

The initial grain size has been missing for a lot of cited literature, such as line 102, line 164. Also, lots of these information can just be summarised in a Table, it is a more clear way to show the results. While those in the main text should have more discussion, not just show the basic results.

Most of the figures are not quite necessary, they just show the basic results of the change of grain size, as commented before, these are basic information which can be summarised in the Table, the texts which only describe those basic results are not quite necessary as well.

Line 45, ‘Many methods have been applied to grain refinement, such as metal forming, and severe plastic deformation’. Justifications are needed, Consider: Manufacturing a curved profile with fine grains and high strength by differential velocity sideways extrusion; A comparative study on deformation mechanisms, microstructures and mechanical properties of wide thin-ribbed sections formed by sideways and forward extrusion

The effect of temperature on the twins has not been thoroughly discussed, for HCP alloys such as Mg.

What is Fig. 25 for? Discuss the difference between 250C and 300C.

Round 2

Reviewer 1 Report

Comments have been addressed well.